Ranking of critical species to preserve the functionality of mutualistic networks using the k-core decomposition

García-Algarra Javier 1 2
Pastor Juan Manuel 2 3
Iriondo José María 4
Galeano Javier javier.galeano@upm.es 2 3
1 Centro Universitario U-TAD , Las Rozas , Spain
2 Complex Systems Group, Universidad Politécnica de Madrid , Madrid , Spain
3 E.T.S.I.A.A.B., Universidad Politécnica de Madrid , Madrid , Spain
4 Area of Biodiversity and Conservation, Universidad Rey Juan Carlos , Móstoles , Spain
Traveset Anna
Electronic publication date: 2017 May 18
Publication date: 2017
Volume: 5
Electronic Location ID: e3321
Received 2017 Mar 7; Accepted 2017 Apr 15
Copyright: ©2017 García-Algarra et al.
Copyright year: 2017
Copyright holder: García-Algarra et al.
License: This is an open access article distributed under the terms of the Creative Commons Attribution License, which permits unrestricted use, distribution, reproduction and adaptation in any medium and for any purpose provided that it is properly attributed. For attribution, the original author(s), title, publication source (PeerJ) and either DOI or URL of the article must be cited.
License URL: https://creativecommons.org/licenses/by/4.0/

Keywords: Mutualism, k-core decomposition, Robustness, Complex networks

Funding: Ministry of Economy and Competitiveness of Spain MTM2012-39101 MTM2015-63914-P This work was supported by the Ministry of Economy and Competitiveness of Spain (MTM2012-39101, MTM2015-63914-P). There was no additional external funding received for this study. The funders had no role in study design, data collection and analysis, decision to publish, or preparation of the manuscript.

==============================
Background

Network analysis has become a relevant approach to analyze cascading species extinctions resulting from perturbations on mutualistic interactions as a result of environmental change. In this context, it is essential to be able to point out key species, whose stability would prevent cascading extinctions, and the consequent loss of ecosystem function. In this study, we aim to explain how the k-core decomposition sheds light on the understanding the robustness of bipartite mutualistic networks.

Methods

We defined three k-magnitudes based on the k-core decomposition: k-radius, k-degree, and k-risk. The first one, k-radius, quantifies the distance from a node to the innermost shell of the partner guild, while k-degree provides a measure of centrality in the k-shell based decomposition. k-risk is a way to measure the vulnerability of a network to the loss of a particular species. Using these magnitudes we analyzed 89 mutualistic networks involving plant pollinators or seed dispersers. Two static extinction procedures were implemented in which k-degree and k-risk were compared against other commonly used ranking indexes, as for example MusRank, explained in detail in Material and Methods.

Results

When extinctions take place in both guilds, k-risk is the best ranking index if the goal is to identify the key species to preserve the giant component. When species are removed only in the primary class and cascading extinctions are measured in the secondary class, the most effective ranking index to identify the key species to preserve the giant component is k-degree. However, MusRank index was more effective when the goal is to identify the key species to preserve the greatest species richness in the second class.

Discussion

The k-core decomposition offers a new topological view of the structure of mutualistic networks. The new k-radius, k-degree and k-risk magnitudes take advantage of its properties and provide new insight into the structure of mutualistic networks. The k-risk and k-degree ranking indexes are especially effective approaches to identify key species to preserve when conservation practitioners focus on the preservation of ecosystem functionality over species richness.

Introduction

Biotic interaction networks play an essential role in the stability of ecosystems (Tylianakis et al., 2010), as well as in the maintenance of biodiversity (Bascompte, Jordano & Olesen, 2006). Because community dynamics greatly depend on the way species interact, these networks have been described as the “biodiversity architecture” (Bascompte & Jordano, 2007). Network analysis has become an important approach to provide information on community organization and to predict dynamics and species extinctions in response to ecosystem disturbance (Tylianakis et al., 2010; Thébault & Fontaine, 2010; Traveset & Richardson, 2014). Among other assessments, these studies can point out key species, whose stability would prevent cascading extinctions, and the consequent loss of biodiversity (Sole & Montoya, 2001; Suweis et al., 2013; Dakos et al., 2014; Santamaría et al., 2015). Research on cascading species extinctions as a result of perturbations in biotic interactions has tackled two main issues: the different ways to rank a hypothetical extinction sequence and the robustness and fragility measures (Pocock, Evans & Memmott, 2012; Domínguez-García & Muñoz, 2015). There are different strategies both to sort species according to their importance and to measure their influence on extinction. For instance, in early studies on the resilience of food webs Dunne, Williams & Martinez (2002) ranked species by degree (i.e., the number of interactions) using three different scenarios of removal: (a) from the species with the highest degree to the species with the lowest degree; (b) from the lowest to the highest; (c) species selected in a random way. Memmott, Waser & Price (2004) worked the same idea to assess the robustness of mutualistic communities, removing active species (in this context, pollinators or seed dispersers) and measuring the fraction of remaining passive species (plants).

An observed property of mutualistic interactions is the existence of generalists, highly interconnected, and specialists, with few interactions linked to the generalists, but rarely among them. The nucleus of interactions among generalists seems to be the foundation of resilience. This property has been traditionally identified with nestedness (Bascompte et al., 2003), although not all mutualistic communities are nested (Joppa et al., 2010; Staniczenko, Kopp & Allesina, 2013). There are new approaches to describe this structure in a more general way as a core–periphery organization (Csermely et al., 2013; Rombach et al., 2014). The core is the set of central and densely interconnected nodes. Ties of periphery are sparse and usually with nodes of the core.

Identification of key species for community preservation is another active field of research. Besides classical measures of node centrality such as closeness, degree or betweenness (Callaway et al., 2000), new rankings based on the well-known Google’s PageRankTM algorithm are now available for ecological networks (Allesina & Pascual, 2009). There are efficient ways to find out these key species in bipartite networks that have been tested in one of the static extinction scenarios we use in this paper (Tacchella et al., 2012; Domínguez-García & Muñoz, 2015).

Another scenario to identify the key species relative to the role species play in the architecture of the network is the study of modularity (Guimerà & Amaral, 2005; Blondel et al., 2008; Guimerà & Sales-Pardo, 2009). A module is a group of species more closely connected to each other than to species in other modules. This ideas have been used in the study of mutualistic networks, estimating how much modular mutualistic network are (Olesen et al., 2007). This analysis allows classifying the nodes into different roles but does not provide a species ranking for possible extinction scenarios. Furthermore, we focus on the relationship of the species with the innermost core of the network, because this core is the cornerstone to understand the functionality of the network.

In this paper, we aim to explain how the k-core decomposition, sheds light on the understanding of robustness in mutualistic networks. The tool classifies the nodes of the network in shells, as in an onion-like structure with the most connected nodes in its center. Taking into account these basic topological properties, the decomposition helps to assess in detail the structure of mutualism and enlightens on the processes of species extinction cascades. Derived from the k-core decomposition we introduce three new magnitudes, hereafter called k-magnitudes, that describe network compactness, defined as the connection to the innermost shell of the network, (k-radius), combined quantity and quality of interactions (k-degree) and species vulnerability to trigger extinction cascades (k-risk). We assess the best criteria for identifying the species for which the networks are most vulnerable to cascade extinctions by comparing k-degree and k-risk ranking criteria with ranking by well-known indexes and applying them in two network destruction procedures. To conduct the test, we use one of the most complete available data sets (Fortuna, Ortega & Bascompte, 2014).

Materials and Methods

Data

We have analyzed the Web of life collection (Fortuna, Ortega & Bascompte, 2014), comprised by 89 mutualistic networks, with 59 communities of plants and pollinators and 30 of seed dispersers (http://www.web-of-life.es/). There are 57 communities with binary adjacency matrix (i.e., the interaction between the two species is recorded but not its strength), and 32 with weighted matrix, where the strength is accounted for. Network sizes range from 6 to 997 species, the minimum number of links is 6 and the maximum is 2,933.

Decomposition and k-magnitudes

The idea of core decomposition was first described by Seidman (1983) to measure local density and cohesion in social graphs. It has been successfully applied to visualize large systems and networks (Alvarez-Hamelin et al., 2005; Kitsak et al., 2010; Zhang et al., 2010; Barberá et al., 2015).

The k-core of a network is a maximal connected sub-network of degree greater or equal than k. That means that each node in the sub-network is tied to at least k other nodes in the same sub-network.

A simple algorithm to perform the k-core decomposition prunes links of nodes of degree equal or less than k (Batagelj & Zaversnik, 2003). The process starts removing links with one of their edges in a node of degree 1. This procedure is recursive and ends when all the remaining nodes have at least two links. The isolated nodes are the 1-shell. Then it continues with k = 2, and so on. After performing the k-decomposition, each species belongs to one of the k-shells (Fig. 1). The m-core includes all nodes of m-shell, m + 1-shell... with m ranging from 1 to the max k index of that particular network. For instance, the 2-core of Fig. 1 is the union of the 1- shell and the 2-shell.

Figure 1 k-core decomposition of a fictional network.

Green nodes are pruned during the first iteration, orange during the second and blue during the last one.

Mutualistic networks are bipartite, with two guilds of species (plant–pollinator or plant-seed disperser in the studied collection). Links among nodes of the same class are forbidden. We will call these guilds A and B.

Based on the k-core decomposition, we define three k-magnitudes. In order to quantify the distance from a node to the innermost shell of the partner guild, we define kradius. The kradius of node m of guild A is the average distance to all species of the innermost shell of guild B. We call this set NB. (1) kradiusAm=1∣NB∣∑j∈NBdistmjm∈A,∀j∈B

where distmj is the shortest path from species m to each of the j species that belong to NB. The minimum possible kradius value is 1 for one node of the innermost shell directly linked to each one of the innermost shell set of the opposite guild.

To obtain a measure of centrality in this k-shell based decomposition, we define kdegree as (2) kdegreeAm= ∑jamjkradiusBjm∈A,∀j∈B

where amj is the element of the interaction matrix that represents the link, considered as binary. If the network is weighted, amj will count as 1 for this purpose if there is interaction, 0 otherwise. It could be understood that kdegree(m) is a fine-grained measure, whereas degree would be a coarse-grained measure. kdegree(m) is like a “continuous” degree where each node i linked to node m adds the inverse of its kradius(j). Generalists score high kdegree, whereas specialists, which have only one or two links, with similar kradius, score lower kdegree. This magnitude reminds the definition of the Harary index (Plavšić et al., 1993) but only considering paths from the nodes tied from m to the nodes of the innermost shell.

Figure 2 shows how kdegree works for one particular network. There are many nodes with the same degree value (Fig. 2A), such as specialists with just one or two links, that from a ranking point of view are equivalent. On the contrary kdegree, maps the degree distribution onto a more continuous one (Fig. 2B), because of the contribution of the inverse of kradius. In Fig. 2C the cumulative distributions of both indexes are overimposed over the degree scale (We perform the linear regression kdegree = α × degree and fit the kdegree distribution to the degree values).

Figure 2 (A) Degree, (B) kdegree and (C) overimposed kdegree distributions of a big plant pollinator community in Central Los Andes, Chile (Arroyo, Primack & Armesto, 1982).

Finally, we introduce krisk as a way to measure how vulnerable is a network to the loss of a particular species: (3) kriskAm= ∑jamjkshellAm−k shellBj+ϵk shellAmm∈A,∀j∈B,k shellBj<k shellAm.

The krisk of a given species is the sum of the k-shell differences to all the nodes of the other guild on a lower k-shells to which it is connected. Each one is weighted by the difference of the k indexes. The second element of Eq. (3) is meant to solve ties among species when they belong to different k-shells, and is a very small quantity (in our implementation we use 0.01, two orders of magnitude lesser than the sum).

In an intuitive way, if we remove one node strongly connected to others of lower k-shells, these species are in high risk of being dragged by the primary extinction. On the other hand, the extinction is much less dangerous for the species of higher k-shells linked to the same node, because they enjoy more redundant paths towards the network nucleus.

Applying the k-magnitudes to a network

Figure 3 is an small seed disperser network with five species of plants, four species of thrushes and eleven links. We call, by convention, guild A the set of plants, and guild B the set of birds. The k-core decomposition was performed with the R igraph package (Csardi & Nepusz, 2006). The maximum k index is 2. The four bird species belong to 2-shell; there are three plant species in 1-shell and two in 2-shell. In this example each species of 2-shell is directly tied to all species of the opposite guild 2-shell, but this is not a general rule.

Figure 3 Computation of the k-magnitudes.

Seed disperser network in Santa Bárbara, Sierra de Baza (Spain) (Jordano, 1993). (A) Decomposed network. (B) Computing kradiusA4.

The shortest path from plant species 2 to each of the four bird species of 2-shell is 1, because of the direct links. So, kradiusA2 is 1. The same reasoning is valid for plant species 1. The reader may check that the kradius of bird species of 2-shell is 1 as well, measuring their shortest paths to plants species 1 and 2.

Computation of this magnitude is simple although a bit more laborious for 1-shell plant species. We work plant species 4 as an example. First, we find the shortest paths to each bird species of 2-shell. Shortest paths are depicted with different colors. Plant species 4 is tied to seed disperser species 1, so distance is 1. On the other hand, there is no direct link with bird species 2. Shortest path is pl4-disp1-pl2-disp2, and distance is 3. It is easy to check that distances from plant species 4 to bird species 3 and 4 are also 3. Once we have found the four distances, we compute kradiusB4 as the average of 1, 3, 3 and 3, that is 2.5.

The values of kdegree are straightforward to compute. For instance, the kdegree of disperser species 1 is: (4) kdegreeB1=1kradiusA1+1kradiusA2+1kradiusA4+1kradiusA5=2.8.

The last k-magnitude we defined was krisk. We use again the disperser species 1 as example. Links to species of the same or upper k-shells are irrelevant to compute krisk, so only plant species 4 and 5 are taken into account. (5) kriskB1=k shellB1−k shellA4+k shellB1−k shellA5+ϵk shellB1=2−1+2−1+0.01x2=2.02.

This magnitude may seem counter-intuitive, because the krisk of a highly connected species like plant 1 is 0.02, almost the same of that of peripheral plant 3 (0.01). This is because plant 1 has no ties with lower k-shell animal species. The krisk ranks species to assess resilience, it has not an absolute meaning. It just tells us that it is more dangerous for the network to remove the disperser 1 than plant 1, and plant 1 than plant 3 (Table 1).

Table 1 K-magnitudes of the network of Fig. 3.

Species	kshell	kradius	kdegree	Rank by krisk	
pl1	2	1	4	3rd	
pl2	2	1	4	3rd	
pl3	1	2.5	1	4th	
pl4	1	2.5	1	4th	
pl5	1	2.5	1	4th	
disp1	2	1	2.8	1st	
disp2	2	1	2.4	2nd	
disp3	2	1	2	3rd	
disp4	2	1	2	3rd	

The k-magnitudes of the example network are shown in Table 1.

Extinction procedures

In order to rank the critical species to preserve the functionality of mutualistic networks and visualize eventual decomposition of the giant component (i.e., the highest connected component of a given network), we carried out two static extinction procedures. Static assumption implies that there is not rewiring (e.g., plants that have lost their pollinators are not pollinated by other insects), despite this kind of network reorganization is observed in nature (Ramos-Jiliberto et al., 2012; Goldstein & Zych, 2016; Timóteo et al., 2016). Nodes are ranked once, before the procedure starts, as in most of robustness assessments studies (Memmott, Waser & Price, 2004; Kaiser-Bunbury et al., 2010; Domínguez-García & Muñoz, 2015).

In the first method, one species is removed each step, in decreasing order according to the chosen index, no matter to which guild it belongs. Four ranking indexes are compared: krisk, kdegree, degree and eigenvector centrality. The k indexes were computed with the R package kcorebip; degree and eigenvector centrality with the degree and evcent functions of the igraph package.

To estimate the damage caused to the network, the fraction of remaining giant component was used. The procedure stops when this ratio is equal or less than 0.5. To break ties, we ran 100 experiments for each network and index, shuffling species with the same ranking value. The percentage of removed species needed to get to 0.5 of the remaining giant component is used to measure the performance of the ranking. The lower the percentage of removed species, the more efficient the ranking is in destroying the network. The top performer scores the least average removal percentage (Fig. 4).

Figure 4 Results of the first extinction procedure for the pollinator network number 10 of the web of life collection.

Performance of the four ranking indexes for a pollinator community described by Elberling and Olesen in Zackenberg Station (Greenland). Individual dots are the results of each experiment while black dots are the average values. The horizontal dispersion is just added jitter for visualization. Performance plots of the 89 networks are available at the github repository, more details in the additional information subsection.

The second extinction procedure that we followed is more common in the literature. Only animal species are actively removed (primary extinctions); secondary extinctions happen when nodes become isolated (Memmott, Waser & Price, 2004).

The fraction of surviving plant species is measured as a function of the removed fraction of animal species (Figs. 5A and 5C) and the area under the curve is the value to compare performance. We averaged the results of 100 repetitions.

Figure 5 Extinction curves of the second algorithm for the pollination network network number 07 of the collection, Suffolk, UK (Dicks, Corbet & Pywell, 2002).

(A, C) Percentage of surviving giant component (GC) and percentage of surviving plant species removing animal species ranked by MusRank. (B, D) Percentage of surviving giant component (GC) and percentage of surviving plant species removing animal species ranked by kdegree. AUCs of the 89 networks are available at the github repository, more details in the additional information subsection.

In this case, in addition to the four indexes of the first experiment, we include MusRank a non-linear ranking algorithm for bipartite networks (Tacchella et al., 2012), inspired by PageRank (Allesina & Pascual, 2009). This algorithm is not valid for the first extinction method. Domínguez-García & Muñoz (2015) showed that MusRank achieves excellent performance for this extinction procedure.

In the second extinction procedure, we also measured the fraction of remaining giant component (Figs. 5B and 5D). Extinction sequences are identical, the only difference is that both magnitudes are measured for each step.

Results

First extinction method

krisk was the ranking method with the lowest average species removal percentage to destroy half of the Giant Component in most of the networks (67 out of 89 networks) (see supplemental material, Table S1). Figure 6 shows the performance comparison of the four ranking criteria. There are some ties, more frequent when networks are small. Network size is the key factor to explain why the performance range is so wide.

Figure 6 First extinction method results.

The average percentage of removed species to destroy the Giant Component (GC) is depicted for each network and ranking index. Under the X axis, the name of each network as coded in the web of life database. The overall top performer is krisk (see Table S1). Species are ordered by the percentage of primary extinctions, ranked by krisk. The red line joins the krisk destruction percentage values as a visual reference to compare them with those of the other indexes.

As size increases, the removal percentage to break the giant component decreases (Fig. 7). When the network is big, the primary extinction of key nodes triggers an important amount of secondary ones. If the community has 100 or more species, krisk is even a better predictor of the most damaging extinction sequence and outperforms the other indexes for 28 out of 32 networks.

Figure 7 First extinction procedure.

Average percentage of removed species to destroy the Giant Component of the top performer ranking as a function of the network size. Dots represent the best result for each network, when there are ties among several rankings for the same network, they overlap.

Second extinction method

MusRank ranking method had the lowest area under the extinction curve for 85 of the 89 studied networks (Fig. 8), and in the other 4 the difference is so small that may be just an effect of the averaging procedure. So, MusRank is the optimal ranking index to destroy the network following this algorithm.

Figure 8 Second extinction procedure, area under the curve of the surviving fraction of plant species as a function of the fraction of removed animal species.

AUCs are plotted for each network and ranking. The overall top performer is MusRank (see Table S2). The solid line joins the MusRank values. Species are ordered by the percentage of primary extinctions, ranked by MusRank.

On the contrary, when the efficiency of the network destruction was measured through the area under the curve (AUC) of the surviving Giant Component fraction the MusRank index had the highest values, placing it as the least efficient ranking method according to this criterion (Fig. 9). In this case, kdegree is the most efficient index for 42 out of 89 networks. We must underline that the extinction sequences are the same, the only difference is the measured output.

Figure 9 Second extinction procedure, area under the curve of the surviving fraction of the original size of the Giant Component as a function of the fraction of removed animal species.

AUCs are plotted for each network and ranking. The overall top performer is kdegree (see Table S3). The solid line joins the kdegree values. Species are ordered by the percentage of primary extinctions, ranked by kdegree.

We have worked out one example (Fig. 5) to explain this shocking difference in performance depending on the measured outcome. On the upper row (subplots A and B), the difference for both ranking indexes when measuring the giant component. While this magnitude decreases at a constant pace for MusRank, there is a sharp reduction of the component size when one third of animal species are removed following the kdegree ranking. On the lower row (subplots C and D), opposite results are obtained when accounting for the fraction of surviving plant species.

The destruction of this pollinator network sheds light on the root cause of the difference. The network has 36 pollinator and 16 plant species (Fig. 10A), two of them are outside the giant component. When the 13 top animal species ranked by MusRank are removed (pollinators 3, 1, 7, 15, 32, 6, 14, 33, 13, 31, 8, 16, 10), the community reaches the degraded structure of Fig. 10B. The size of the giant component is 27 (54% of the original), and there are 23 pollinator and 6 plant species.

Figure 10 Pollinator network 007 (Dicks, Corbet & Pywell, 2002).

(A) original configuration; (B) structure after the removal of the 13 top MusRank-ranked animal species. (C) structure after the removal of the 13 top kdegree-ranked animal species.

If we remove the 13 top animal species ranked by kdegree (pollinators 1, 3, 7, 13, 15, 2, 11, 20, 12, 8, 6, 5, 10) instead, the community structure is that of Fig. 10C. Now, the size of the giant component is 19 (38% of the original), and there are 23 pollinator and 9 plant species. MusRank has killed more plant species, but the giant component is clearly smaller ranking by kdegree.

Discussion

The k-core decomposition offers a new topological view of the structure of mutualistic networks. We have defined three new magnitudes to take advantage of their properties. Network compactness is described by kradius, a measure of average proximity to top generalists of the partner guild. Second, kdegree maps each node’s degree onto a finer grain distribution. It has not only information on the number of neighbors but also on how they are connected to the innermost shell. Finally, krisk is set to identify species whose disappearance poses a greater risk to the entire network.

Comparing the k-magnitudes based extinction indexes (kdegree and krisk) with those routinely used when extinctions take place in both guilds, krisk is the best rank if the goal is to identify the key species to preserve most of the giant component. krisk identifies species linked to a high number of nodes of lower k-shells. These species provide vulnerability to the network because their extinction may drag many of the species with lower k-shells they are linked to, to extinction as well, as they do not enjoy redundant paths to the innermost shell.

Applying the well-known method of removing species of the primary class and measuring the extinctions in secondary class, the most effective extinction sequence, if the goal is to identify the key species to preserve most of the giant component, is kdegree. However, if the goal is to identify the key species to preserve the greatest species richness in the second class (e.g., plants in a plant–pollinator mutualistic network), the best criterion is MusRank as Fig. 8 makes clear. These results confirm those obtained by Domínguez-García & Muñoz (2015), over a larger network collection (89 in this work vs. 67 in the original paper).

The most striking result of the second method is how different performance is for a same ranking index, depending on the magnitude we measure. The root cause lies on the definitions of the indexes themselves. MusRank is optimal to destroy the plant guild. It identifies the most important active nodes of the bipartite network because of how they are linked to the most vulnerable passive ones. It was designed to excel with this extinction sequence and works with local properties. On the other hand, kdegree is an excellent performer to destroy the giant component. It contains information on how nodes are connected to the innermost shell, and ranks higher those nodes strongly tied to that stable nucleus. This new approach to network robustness could be also applied to other types of networks in which the integrity of the giant component is more important than the number of remaining nodes, for example in communications or epidemic spread networks.

In summary, in this study, we show that the new k-core decomposition derived indexes, krisk and kdegree provide a new insight into the structure of mutualistic networks. This insight is particularly useful because these indexes fair much better than other traditionally used ranking indexes, when the aim is to identify the species that are key to preserving the interactions and the functionality of the community. As complex network studies on mutualistic interactions are already being used to suggest conservation policies, it is of utmost importance to have a clear framework of what the conservation practitioners look for when implementing conservation and restoration plans. The static view of considering biodiversity conservation as the mere conservation of a list of species has long been substituted by a new paradigm which looks at conservation from a dynamic viewpoint in which species interactions and the functionality of the ecosystems play a major role (Heywood & Iriondo, 2003).

Supplemental Information

Supplemental Information 1 Average number of removed species

Click here for additional data file.

Additional Information and Declarations

Competing Interests

Author Contributions

Data Availability

The authors declare there are no competing interests.

Javier García-Algarra conceived and designed the experiments, performed the experiments, analyzed the data, contributed reagents/materials/analysis tools, wrote the paper, prepared figures and/or tables, reviewed drafts of the paper, wrote the R-code.

Juan Manuel Pastor conceived and designed the experiments, performed the experiments, analyzed the data, contributed reagents/materials/analysis tools, wrote the paper, prepared figures and/or tables, reviewed drafts of the paper, wrote the Python code.

José María Iriondo conceived and designed the experiments, wrote the paper, reviewed drafts of the paper, ecological interpretation.

Javier Galeano conceived and designed the experiments, performed the experiments, analyzed the data, wrote the paper, reviewed drafts of the paper.

The following information was supplied regarding data availability:

The R code for k-core decomposition and plotting has been published as a package at https://zenodo.org/badge/latestdoi/58714207.

The rest of the software is available at https://zenodo.org/badge/latestdoi/78195863.

Reproducibility instructions are detailed in the README.md file.

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
