# Peer review of "Ranking of critical species to preserve the functionality of mutualistic networks using the k-core decomposition"

_PeerJ, doi:10.7717/peerj.3321_

## Round 0.1 · original submission · Minor Revisions

· Academic Editor

Minor Revisions

Your paper has been reviewed by two experts on networks who are very positive about it. Both referees make useful suggestions that I think you should consider to improve the current version of the paper.
I agree with ref 2 that it would be good if you could clearly state why the k-core analysis improves the classification in modules given by other analysis such as Netcarto (or other software providing modules).
I look forward receiving your revised ms in the next few weeks.

Sincerely,
Anna Traveset

·

Basic reporting

The paper reports clearly a new technic and metrics to explore bipartite networks. I have a few suggestions on how to clarify some passages, specially for empiricists that want to get the gist without going through too much mathematical jargon (see below).

Experimental design

The methods are clear and reproducible and tested on large dataset of available plant-pollinator networks.

Validity of the findings

The paper describes a new methodology and tested it in a large number of networks. Data is robust but the results are most times descriptive. I am ok with showing the rankings of which metric ranks best for each extinction method instead of formally testing if is significantly better in a frequentist framework, however, some readers may want to see this comparison explicit. Some statements like in line 170 mentioning that network size is correlated with species needed to break the giant component may be more convincing if formally tested.

Additional comments

This is a very interesting paper introducing more metrics to the ever-growing network metric lists. I have a set of suggestions to make the paper more accessible to the potential users and ask for some clarifications.

line 15: The first line is a bit vague to me. Ecological communities are, in fact, the biodiversity we want to preserve in first place, so of course they play a role.

line 16: measurements of "species" centrality or degree.

line 16: The abstract in general is very short and if space allows you can expand for example indicating why are they modeled as bipartite networks.

Line 38: Active and passive are terms that may need a brief explanation in this context.

line 45: This paragraph needs a concluding sentence. What’s going on with this approach, does it have limitations? How it finds the core? How it compares to your K-core decomposition?

Line 47: Explain centrality briefly maybe?

Line 48. Same as above. Which are this new rankings? Do they have limitations?

Line 50: In mutualistic networks (not mutualism)

Line 54: I am not familiar with the term compactness. What this refer too? Can be related to nestedness and how we identify the core of species?

Line 72: each node "in the sub-network"? is tied to…

Line 77: Here you introduce the subscript m. What this means?

Line 84: I know is easy to see from the very nice examples, but you can define what you mean by distance.

Eq (1) in eq.2 m exists in A and j exists in B. Should be the same constraint for Eq A?

Line 91: This explanation “weighted degree” is confusing because you are using binary networks. I see you weight for the partners degree, but can you find a better naming, given other weighted degree metrics use the strength of the links as weights?

Line 91: you mean node j? I think so, according to the equation. Clarify.

Line 98: The degree “of the partners”?

line 99: Define degree scale even if briefly.

Eq3 : idem, unify i and j notations if possible. I think you use here i, but above the same concept is expressed as j. Also i “in” B should be the symbol for “exists in” as above.

Fig 3: Can you color code the numbers in the equation at the bottom of B so they match the colors of the boxes? This will facilitate linking them to the figure. In caption, explain colors.

Line 129: give value of plant 3 in text (0.01 if I am correct).

Line 133: This is an important point. Giving that the numerical differences are meaningless, and the ranking is the important thing, I think is confusing to keep the actual numbers. I suggest transforming this value to a ranking in a final step and do not show the intermediate calculations explicitly, which people will try to interpret and are very tricky to interpret correctly. This is important, specially in the output of the package.

Line 135-138: You may want to sell why this approach is useful. Now you only mention it ignores rewiring letting the reader wondering if it makes any sense to use it then. (I do thing it makes sense depending on purpose, but you need to convince the reader).

Fig 5: Explain in caption that GC is “giant component”.

Table 2, 3 and 4: Those are hard to read tables. I suggest trying to plot some of this info and move the rest to sup mat.

line 199: same comment for compactness. Can you relate k-degree to contribution to nestedness core, for example?

Discussion point: Can you discuss how this is potentially useful in other contexts. E.g. information spread in the network (keystone species, or "hubs"), other robustness metrics attacking individuals or links, and not species.

Congratulations for releasing the code, but github is not a permanent repository. You can easily attach a DOI to the github repo and make that permanent, for example using Zenodo.

Best,
Ignasi Bartomeus

·

Basic reporting

- The manuscript is well written, with a good and professional level of English being used. The text is mostly clear and unambiguous throughout, with a few minor imprecisions that do not impend the understanding of the main message. These are given in the comments to the authors box.

- The authors provide a generally good contextualization of their work, with enough and relevant references. They start out explaining the use of network theory to the study of biotic interaction, how they can help to understand community organization, and what it might mean in terms of the dynamics of these systems and in terms of extinction processes. The authors then introduce the subject of the classification of the importance of species in a network, and how they influence extinctions cascades.

- Figures are appropriate and clear, though Figure 6 is slightly “eye straining”. The vertical dashed lines and the clumped labels with the networks names on the horizontal axis make it hard to read. If this one could match the clarity of the similar figures 7 and 8, it would be great. I don’t think tables 2, 3, and 4 are really necessary, and the information given there is already displayed in figures 6, 7, and 8.

- Raw data are available.

Experimental design

- The authors explained the tools they are going to apply (k-decomposition), and its basic rational, and introduced the indices they derived (the “k-magnitudes”) and how they work.

- The question at the heart of this work is clearly stated (“identify the species for which networks are more vulnerable”), and the authors identify how they are going to address the suitability of the tools they are applying to that end (performance of the various indices as a criterion to extinction procedures). Furthermore, they compared the performance of these indices against that of other commonly used criteria and indices. The approach present by the authors adds to to the body of tools being transfered from the field of network science to the biology/ecology world.

- The methods presented are clear. The authors provide little examples that help to understand the calculations and rationale of the indices that are being presented. There are a few imprecisions that I detail in the comments for the authors box.

- The code and files necessary to run the analysis are also available, and instructions to follow step by step the procedures are included.

Validity of the findings

- The data is appropriated to the task the authors undertook, with the plus of such data coming from empirical, and real, natural systems, rather than as is seen often artificial computer generated data. The authors make use of an extensive number of data sets publicly available. This is a clear example of the good that may come from publicly sharing data!

- The discussion is tight and to the point, explaining the results obtained and the implication to the questions stated initially by the authors.

Additional comments

Line 46-48 - Ultimately the identification of key species can relate to the role species play in the architecture of a network. Community finding algorithms have been widely applied in the identifying groups of highly connected species -modules - that are more likely to interact with species in the same module than with species belonging to other modules (e.g. Guimerà & Amaral 2008: doi:10.1088/1742-5468/2005/02/P02001; Blondel et al 2008: doi:10.1088/1742-5468/2008/10/P10008). In Guimerà & Amaral 2005 (doi:10.1038/nature03288) they suggest a method to species roles in a network. This has been widely used in the study of mutualistic network, and other kinds of ecological networks, and estimate how much the structure of a network adheres to a modular structure. As such I think it deserves a mention in the present work. What could the k-core analysis improve the classification given by that method?

Line 102 - The explanation of eq. 3 is somewhat imprecise, or at least not complete. What is summed is the differences in k-shell, isn’t it? Would be more correct if it was: "The k-risk of a given species is the sum of the k-shell differences to all the nodes of the other guild on a lower k-shell to which it is connected"? Or something along these lines.

Lines 103-104: I am not sure that I understand the need to solve ties between species, hence the need of the second element in the equation. Is this a mathematical need, or is there a biological reason beyond it? I reckon that though two species may belong to different k-shells their removal could eventually have a similar effect on the vulnerability of the network. In that scenario, for some reason, other than the shell to which a species belong, may determine the end effect of its removal.

Minor comments:
Line 31-32 - “the different ways to rank a hypothetical extinction sequence”: Perhaps a reference could be added exemplifying such attempts to rank extinction sequences.

Line 51 - Which are the very basic topological properties? Or did you mean to say "Taking into account these very basic topological properties(...)"? Please clarify.

Line 97-98 - I think there should be a reference to Figure 2B in this sentence. Otherwise that plot is not mentioned in the text at any time.

Figure 3 (legend) - If I understood well the notation used by the authors, here the superscript should be A. It is the radius of plant4 that is being calculated, isn't it?

Equation 4 – After the eq. 4 the authors exemplify the calculation of k-risk for disperser1 (shown in eq. 5), and say “only bird species 4 and 5 are taken into account”. But should it say “only plant species 4 and 5 are taken into account”? After all, there is no bird species 5!

Line 130-132 - Perhaps a reference to Table 1 should be also placed here, as for the reader would be easier to know where the values discussed here come from.

Figure 4 - I wonder why the individual dots for the extinction procedure using k-risk are spilt so tightly between those two levels.

Line 183-184 - This sentence seems a bit confusing, or at least not very well connected with the surrounding sentences.

---

## Round 0.2 · accepted · Accept

· Academic Editor

Accept

Dear Javier,

Thank you for sending your revised ms after considering the comments by the two referees. I think it has improved a lot and it is now ready for publication.

I would only like you to suggest two very minor things for the abstract:

(1) in the last sentence of the first paragraph (Background), delete the ',' after 'core decomposition'
(2) in the results' paragraph, you refer to MusRank index without having previously mentioned what this index is. Thus, please state in Methods that this index is.

Congratulations for your nice paper!
Best wishes,
Anna Traveset